# Inhibition of Orexin/Hypocretin Neurons Ameliorates Elevated Physical Activity and Energy Expenditure in the A53T Mouse Model of Parkinson’s Disease

**DOI:** 10.3390/ijms22020795

**Published:** 2021-01-14

**Authors:** Milos Stanojlovic, Jean Pierre Pallais, Catherine M. Kotz

**Affiliations:** 1Department of Pharmacology, Toxicology and Pharmacy, University of Veterinary Medicine, Bünteweg 17, 30559 Hannover, Germany; 2Integrative Biology and Physiology, University of Minnesota, Minneapolis, 321 Church St SE, Minneapolis, MN 55455, USA; palla058@umn.edu (J.P.P.); kotzx004@umn.edu (C.M.K.); 3Minneapolis VA Health Care System, GRECC, 1 Veterans Dr, Minneapolis, MN 55417, USA

**Keywords:** Parkinson’s disease, orexin, neuromodulation

## Abstract

Aside from the classical motor symptoms, Parkinson’s disease also has various non-classical symptoms. Interestingly, orexin neurons, involved in the regulation of exploratory locomotion, spontaneous physical activity, and energy expenditure, are affected in Parkinson’s. In this study, we hypothesized that Parkinson’s-disease-associated pathology affects orexin neurons and therefore impairs functions they regulate. To test this, we used a transgenic animal model of Parkinson’s, the A53T mouse. We measured body composition, exploratory locomotion, spontaneous physical activity, and energy expenditure. Further, we assessed alpha-synuclein accumulation, inflammation, and astrogliosis. Finally, we hypothesized that chemogenetic inhibition of orexin neurons would ameliorate observed impairments in the A53T mice. We showed that aging in A53T mice was accompanied by reductions in fat mass and increases in exploratory locomotion, spontaneous physical activity, and energy expenditure. We detected the presence of alpha-synuclein accumulations in orexin neurons, increased astrogliosis, and microglial activation. Moreover, loss of inhibitory pre-synaptic terminals and a reduced number of orexin cells were observed in A53T mice. As hypothesized, this chemogenetic intervention mitigated the behavioral disturbances induced by Parkinson’s disease pathology. This study implicates the involvement of orexin in early Parkinson’s-disease-associated impairment of hypothalamic-regulated physiological functions and highlights the importance of orexin neurons in Parkinson’s disease symptomology.

## 1. Introduction

Parkinson’s disease (PD) is a neurodegenerative disease that is accountable for about 50% of all synucleopathies, which are diseases defined by abnormal accumulation of alpha-synuclein (α-syn) aggregates. PD affects 1–2 per 1000 of the population at any time, with 1% of people over the age of 65 and up to 5% of people over the age of 85 [1,2,3], and its prevalence is second only to Alzheimer’s disease. Interestingly, males are about 1.5-times more susceptible to PD and this likelihood increases with age [4]. Classical hallmarks of PD include the presence of Lewy bodies and dopaminergic neuron loss in the substantia nigra which leads to signature movement disorders. However, in recent years, PD has become recognized as a multilayered disease. Neurodegeneration is not exclusive to dopaminergic neurons [3,5], and it is shown that mood, cognition, and metabolic impairments are present even prior to the onset of the hallmark motor impairments [6,7,8,9].

Orexin (hypocretin) is a neurotransmitter simultaneously discovered 20 years ago by two independent groups [10,11]. A distinctive population of neurons expressing orexin has been observed in the lateral hypothalamus (LH), a brain region responsible for many functions including feeding behavior, arousal, and pain perception [12]. This distinctive population of neurons has a complex pattern of projection throughout the brain [13,14], enabling them to regulate many different processes [15,16,17,18,19,20,21,22]. Nevertheless, the orexin system is most known for its role in hypothalamic-regulated physiological functions [23,24,25,26,27,28], including spontaneous physical activity (SPA) and energy expenditure (EE) [29,30,31].

There are several studies that emphasize the role of orexin in PD pathology. One of the major non-motor symptoms of PD is sleep impairment, which can be observed early in PD progression [32,33]. Sleep impairments are associated with orexin system dysfunction and reduced levels of orexin have been detected in the cerebrospinal fluid of PD patients [34,35]. Finally, orexin neuronal loss has been observed in PD patients [36,37], further implicating reductions in orexin tone with PD symptomology.

The human alpha-Syn (A53T) transgenic line G2-3 mice overexpress the mutant form of human α-syn associated with familial PD. In these transgenic mice, expression of the A53T missense mutant form of human α-syn is under the control of the murine prion promoter (Prp). Transgenic A53T mice show the complete α-syn pathology that is observed in humans [38] and is extensively studied in the context of neurodegeneration, α-syn aggregation, and toxicity [39]. These mice spontaneously develop the neurodegenerative disease between 9 and 16 months of age with a progressive motoric dysfunction leading to death within 14–21 days of onset [40]. Overexpression of A53T mutant human α-syn increases neuronal toxicity, inflammation, and astrogliosis, and impairs neuronal function in A53T mice [41,42,43,44,45,46].

Classically, PD is considered a motor disorder driven by dopamine system impairment, but the importance of non-motor symptoms and pathophysiological mechanisms involving different brain regions, neuronal populations, and signaling pathways has been recently recognized [47,48]. In this study, we determined the effects of A53T α-syn associated pathology on aspects of energy metabolism, including body composition, exploratory locomotor activity, SPA, EE, and LH orexin. Further, we investigated if chemogenetic inhibition of orexin neurons can ameliorate impairments observed in A53T mice.

## 2. Results

### 2.1. A53T Mice Have Increased Food Intake, Are Leaner, and Have Increased Exploratory Locomotion

To characterize aging-induced changes in food intake, body mass, composition, and exploratory locomotion we used 3-, 5-, 7-, 9-, and 11-month-old (mo) male WT and A53T mice. For more detailed information regarding food intake, body mass, lean mass, fat mass, fat to lean mass ratio, and distance traveled, please see Table 1. With the exception of 11-month-old A53T mice, food intake increased with age in both WT and A53T mice (Figure 1B). Further, compared to age-matched WT controls, A53T mice consumed more food (Figure 1B).

As expected, body mass increased with age in all mice (Figure 1C). Statistically significant differences in lean body mass were not observed (Figure 1D), however, fat mass was significantly affected by aging in both WT and A53T mice (Figure 1E). Differences in fat mass were observed between WT and A53T mice at 11 months of age (Figure 1E). Fat to lean ratio increased in an age-dependent manner in both WT and A53T mice (Figure 1F). Furthermore, in the A53T mice, the fat to lean ratio was significantly lower compared to WT mice (Figure 1F).

Locomotor activity declined with age in WT mice (Figure 1G). In contrast, exploratory locomotion increased in A53T mice with age, until 11 months (Figure 1G). Finally, differences in exploratory locomotion between WT and A53T mice were observed (Figure 1G).

### 2.2. A53T Mice Have Increased SPA and EE

In the light, inactive phase, SPA was significantly increased in the A53T mice as compared to that in the WT mice (Figure 2F). Differences in SPA during the light phase were observed at different ages of A53T mice as well (Figure 2F). In the dark, active phase, SPA was significantly increased in the A53T mice as compared to that in the WT mice (Figure 2G). Differences in SPA during the dark phase were observed at different ages of A53T mice as well (Figure 2G). Total SPA of the A53T mice was increased relative to SPA of the WT mice (Figure 2H). Differences in total SPA (light + dark phases) were observed in different ages of A53T mice as well (Figure 2H). For more detailed information regarding number of beam breaks and energy expenditure (light phase, dark phase, and total) please see Table 2.

ANCOVA analysis showed significant differences in SPA within the light phase: F(9, 49) = 8.139; *p* < 0.05; the dark phase: F(9, 49) = 12.045, *p* < 0.005; and across the entire 24 h cycle: F(9, 49) = 6.934, *p* < 0.005; EE. Post hoc testing (Sidak’s) showed that in the light, inactive phase, A53T mice had significantly increased EE compared to that of the WT mice (Figure 3F). In the dark, active phase, EE was significantly increased in the A53T mice as compared to that in the WT mice (Figure 3G). Differences in EE were observed in different age groups of A53T mice as well (Figure 3G), and total EE of the A53T mice was increased with age (Figure 3H).

### 2.3. PD-Associated A53T Pathology Increases Astrogliosis and Inflammation Markers in the LH of A53T Mice in Early Stages of the Disease, and Induces Orexin Neuron Loss in Later Stages of the Disease

P-α-syn aggregations were observed in orexin neurons of 7-month-old A53T mice, and the majority of these aggregations were localized in the nuclei of orexin neurons. Although 74.9 ± 3.9 (mean ± SEM; Figure 4C) of the orexin neurons contained p-α syn aggregations, it did not affect the number of orexin neurons in the LH at 7 months of age (Figure 5H).

Increases in GFAP expression were observed in the LH of 7 mo A53T mice (7 mo WT vs. 7 mo A53T, *** *p* < 0.005; Figure 5E). Increased IBA1 expression in the LH was accompanied by astrogliosis (7 mo WT vs. 7 mo A53T, *** *p* < 0.005; Figure 5F). An increased density of IBA1 positive cells was observed in the LH of A53T mice (7 mo WT vs. 7 mo A53T, *** *p* < 0.005; Figure 5G). Aging-associated reductions in orexin neurons were observed in A53T mice (5 mo A53T vs. 1 mo A53T). There were statistically significant differences in orexin neuron numbers between the WT and A53T mice at 11 months (11 mo WT vs. 11 mo A53T, * *p* < 0.05; Figure 5H).

### 2.4. Loss of GAD65 Presynaptic Terminals in LH of the 7-Month-Old A53T Mice

Quantification of pre-synaptic inhibitory terminals using an immunocytochemistry-based assay showed a reduction in the number of GAD65/synaptophysin co-localizations in A53T compared to WT mice at 7 months of age (7 mo WT vs. 7 mo A53T, ** *p* < 0.01; Figure 6C).

### 2.5. Clozapine N-Oxide (CNO) Treatment in Mice with an Inactive Designer Receptor Exclusively Activated by Designer Drugs (DREADD) Construct Does Not Affect Food and Water Consumption, Exploratory Locomotion, SPA, or EE

Prior to pursuing chemogenetic studies, we addressed a recent report [49] indicating that CNO does not readily cross the blood–brain barrier in vivo. Further, it was reported that CNO converts to clozapine in vivo, which has antipsychotic properties and may affect performance on some behavioral tasks. To exclude potential independent actions of clozapine in the assay readouts, we performed pre-tests in orx-Cre cDREADD and mice to assess if CNO alone affected exploratory locomotion, SPA, and EE. In addition, we determined whether CNO given in the drinking water affected water consumption.

Given acutely, CNO (i.p. injection, 3 mg/kg) did not affect exploratory locomotion (Figure 7A). When dissolved in the drinking water, CNO (0.25 mg/mL) did not affect water consumption (Figure 7B). Mice consumed 5.61 ± 0.34 mL (mean ± SEM) of CNO solution daily, which corresponds to 45.75 ± 2.78 (mean ± SEM) mg/kg of CNO. When administrated via drinking water at 0.25 mg/mL, CNO did not affect food intake, SPA or EE (Figure 7C–E).

### 2.6. Chemogenetic Inhibition of Orexin Neurons Ameliorated Increase in Exploratory Locomotion, SPA, and EE

As expected, exploratory locomotion in orx-Cre/A53T mice was increased compared to that in orx-Cre mice (Figure 8D). Inhibition of orexin neurons using DREADDs reduced exploratory locomotion in orx-Cre/A53T mice (Figure 8A) but did not restore it to levels observed in orx-Cre mice (Figure 8D).

Chemogenetic inhibition of orexin neurons reduced SPA in A53T mice. In the light phase, orx-Cre/A35T mice had increased SPA compared to that in orx-Cre mice (Figure 8E). CNO reduced SPA in A53T mice, without completely ameliorating changes in SPA (Figure 8E). Similar changes were observed in the dark phase (Figure 8F). Orx-Cre/A53T mice showed increased SPA compared to the appropriate controls (Figure 8F). Inhibition of orexin neurons reduced total SPA in orx-Cre/A53T mice (Figure 8G), but did not reduce total SPA of orx-Cre/A53T mice to control levels (Figure 8G).

EE was analyzed in an ANCOVA model with lean body mass as a covariate. ANCOVA analysis showed significant differences in the light phase: F(2, 14) = 5.883, *p* < 0.05; dark phase: F(2, 14) = 74.09, *p* < 0.005; and total: F(2, 14) = 67.703, *p* < 0.005; EE. Sidak’s post hoc test showed that in light phase, compared to control orx-Cre mice, orx-Cre/A53T mice have increased EE (Figure 8H). Chemogenetic inhibition of orexin neurons did not affect EE in the light phase (Figure 8H), whereas in the dark phase increased EE was observed in orx-Cre/A53T mice (Figure 8I). Inhibition of orexin neurons reduced EE in dark phase in orx-Cre/A53T mice (Figure 8I). Total EE was increased in orx-Cre/A53T mice (Figure 8J). Chemogenetic inhibition of orexin neurons ameliorated PD-induced increases in total EE (Figure 8J).

There was a trend towards reduced food intake following chemogenetic inhibition of orexin neurons, but this was not statistically significant (Figure 8K). For more detailed information regarding food intake, distance traveled, number of beam breaks, and energy expenditure (light phase, dark phase, and total), please see Table 3.

### 2.7. Confirmation of Injection Placement and DREADD Functionality

Orx-Cre/A53T mice used in the DREADD study received bilateral DREADD viral injections. Immunohistological analyses confirmed the selective expression of hM3Dq-mCherry in orexin neurons. Clear co-localization of orexin A and mCherry positive cells (OrxA/mCherry, mean ± SEM, cDREADD, 76.35 ± 4.30; Figure 9C) was observed.

## 3. Discussion

PD is a complex, progressive age-associated disease with characteristic symptomology that includes hypokinesia, the presence of abnormal protein particles known as Lewy bodies, and the loss of dopaminergic neurons in the substantia nigra pars compacta. Due to the general aging of the population and expected increase in life expectancy, the PD-associated social and economic burden will significantly increase in the next 25–30 years [50], necessitating new strategies to meet the health care needs of individuals with PD [51]. In recent years the importance of non-motor symptoms and impairments in PD are becoming more recognized as well. Non-dopaminergic neuronal loss [3,5], as well as the presence of non-motor symptoms, such as metabolic impairment, prior to the onset of the disease have been observed [6,52]. There is a rapidly expanding effort to define the prodromal stages of PD, as pre-motor symptomology and pathology in PD is understudied.

The first part of our study addressed aging-induced changes in body composition, exploratory locomotion, SPA, and EE in A53T mice. We observed that A53T mice are leaner and gain less body and fat mass as they age compared to their WT littermates. Further, exploratory locomotion, SPA, and EE observed in A53T mice is significantly higher than that in WT mice, which agrees with earlier studies. Hyperactivity was previously observed in this animal model [53], but this is not the sole factor contributing to reduced fat mass in the A53T mice. Moreover, in our recent, less extensive study, we observed increased exploratory locomotion in A53T mice as well [54]. In a similar transgenic mouse model of PD (A53T expression driven by Thy1 promoter), Rothman et al. [55] showed that metabolic perturbations are present. Namely, mice with A53T expression driven by the Thy1 promoter are hypoleptinemic, have increased EE, and are resistant to high fat diet induced insulin resistance despite consuming more food. The metabolic abnormalities observed in our study resemble those observed in PD patients: weight loss, reduced body fat mass, and increased EE [56,57]. Interestingly, increased SPA and EE in A53T mice was not restricted only to the dark (active) phase of the day. Increases in SPA and EE during the light (inactive) phase can suggest disturbances in sleep regulation, which is a common trait of PD [32,58]. Furthermore, the usual light phase activity pattern is not observed in 11-month-old A53T mice. Unlike the observed energy metabolism impairments, increased exploratory locomotion is less directly in alignment with classical PD symptoms. Several facts may provide a better understanding of the observed phenotype in A53T mice and its relationship to PD. First, A53T-associated pathology leads to dysregulation of the dopaminergic system [53], which leads to elevations in activity similarly to that present in dopamine active transporter knockout animals [59,60]. Second, motor impulsivity (tendency to perform previously learned motor responses despite signals to the contrary) and dis-inhibition are features of PD [61], which could lead to increased locomotion in A53T mice. Finally, this study addresses the early, pre-motor onset phase of the disease, which is an unstudied area.

Exploratory locomotion, SPA, and EE are strongly influenced by orexin system function [28,30]. The orexin receptor 1 (OX1R) antagonist SB334867 reverses chemical vestibular lesion induced hyperactivity, and peripheral injection of JNJ-10397049, a selective OX2R antagonist, decreases ethanol-induced hyperactivity in mice. The dual orexin receptor antagonist almorexant induces sleep and decreases orexin-induced locomotion [62,63,64]. Further, orexin increases SPA while enhancing EE during SPA, rest and sleep [29,65]. Based on these findings, and the observation that there are changes in SPA/EE in A53T mice, we suspected that A53T-associated pathology could be present in in the orexin field within the LH of A53T mice. Previously, we showed the presence of A53T mutant human p-α syn in the LH of A53T mice [66], and, using confocal microscopy, we identified that p-α syn aggregations are localized mostly in the nuclei of orexin neurons. It was also previously established that overexpression of A53T p α-syn in A53T mice increases neuronal toxicity and impairs neuronal function [44,46]. The large detergent-insoluble aggregates of p-α-syn observed in other brain regions of A53T mice are accompanied by mitochondrial degeneration, lysosome pathology, and cell death [67,68]. Furthermore, for the first time to the best of our knowledge, we showed that A53T p-α-syn accumulation-associated inflammation and astrogliosis was present in the LH of A53T mice, which is represented by increased expression of GFAP and IBA1, and increased IBA1 cell counts in 7-month-old mice. Astrogliosis and inflammation are considered major factors in PD [43,45] and are involved in A53T-related pathology [42,45,69,70].

The presence of p-α-syn is considered to be neurotoxic, and thus we were keen to investigate whether this affects orexin neuronal numbers. Using an unbiased stereology approach, significant loss of orexin neurons was determined in 11-month-old A53T mice only. This orexin neuron loss is in agreement with studies demonstrating orexin neuronal loss in PD patients [36,37]. Interestingly, orexin neurons are sensitive to some drugs used in PD treatment. Receptor-mediated presynaptic suppression of glutamatergic excitatory inputs to orexin neurons leads to chronic silencing of orexin neurons and depletion of orexin [71]. This means that orexin neurons are not only vulnerable to PD pathology, but drugs used for PD treatment can also impair their function.

The presence of Lewy bodies in the LH of PD patients is established [69,70] but only one study has addressed if α-syn accumulations are observed in orexin neurons [37]. While data from the study by Thannickal et al., 2007, showed no α-syn accumulations in orexin neurons, more sophisticated approaches could yield a different outcome. Further, alpha-syn accumulations undoubtedly play a role in the pathology of PD, however p-α-syn, insoluble α-syn aggregates, and oligomeric α-syn species are implicated in PD pathophysiology as well [72,73,74]. Moreover, orexin neurons are not the only LH neuronal population susceptible to neurodegeneration, as the loss of LH melanin-concentrating hormone (MCH) neurons has been shown in PD patients [37]. Both orexin and MHC neurons are closely linked to the dopaminergic system [75,76] and it would be interesting to examine if dopamine system impairment is a common underlying mechanism of neurodegeneration in orexin and MHC neurons, or if different mechanisms exist.

A disruption in the balance of the major neurotransmitter levels is a common feature of many neurodegenerative diseases including PD [77,78]. Some have proposed the loss of GABAergic system function as one of the major mechanisms of PD pathology [79,80]. In alignment with this idea, we observed the loss of inhibitory presynaptic terminals to the LH orexin field (as determined by GAD65 staining). It is possible that the loss of inhibitory input to the LH leads to overactivation of orexin neurons, which causes the observed increase in exploratory locomotion, SPA, and EE. Prolonged neuronal overactivation leading to neurodegeneration due to excitotoxicity is a hallmark of different neurodegenerative diseases including PD [81,82]. Orexin neuronal loss could be a consequence of its prolonged overactivation, but more research is needed to address this idea.

Interestingly, sleep impairments such as insomnia, fragmented sleep and narcolepsy are common in PD patients [83]. The orexins play a major role in the regulation of sleep [84] and thus it is not surprising that orexin system dysregulation results in various sleep impairments, including destabilization and loss of REM sleep [34,85,86], more sleep to wake transitions [26], and sleep fragmentation [87]. Perhaps the most interesting link between the orexin system and PD is narcolepsy, which is commonly experienced by PD patients [88]. It is well established that orexin neuronal loss causes narcolepsy [89] and occurs in PD patients [36,37] and thus it is possible that narcolepsy observed in PD is caused by orexin neuronal loss.

The main goal of this study was to determine if in vivo inhibition of orexin neuron activity could ameliorate exploratory locomotion, SPA, and EE disturbances in A53T mice. To achieve this, we used a DREADD approach. Before the main experiment, we addressed concerns related to possible off-target effects of the designer ligand, CNO [49]. The conducted experiments confirmed that none of the CNO treatments (i.p., 3mg/kg; dissolved in drinking water, 0.25 mg/mL) had effects in control mice on exploratory locomotion, SPA, and EE (orx-Cre, cDREADD), mitigating concern over off-target and independent effects of clozapine. To achieve chemogenetic modulation of orexin neurons, we created double transgenic orx-Cre/A53T mice and intracranially injected them with virus containing either control or inhibitory DREADD constructs. After transfection, we subjected them to an experimental procedure assessing exploratory locomotion, SPA, and EE. In our earlier study, we showed that chemogenetic inhibition of orexin neurons ameliorated elevated exploratory locomotion in 5-month-old mice [54]. As hypothesized, we showed that chemogenetic inhibition of orexin neurons ameliorated exploratory locomotion, SPA, and EE impairments present in 7-month-old A53T mice. Not only are orexin neurons involved in control of sleep and wakefulness [25,90], but a recent study established that diurnal reorganization of the excitatory/inhibitory balance in the perisomatic innervation of orexin neurons exists [91]. Simply put, synaptic rearrangement of inputs to orexin neurons happen over the course of the day in relation to sleep and wake states, a phenomenon known as chrono-connectivity. In the current study, differences in orexin-specific neuronal inhibition effects on SPA and EE in A53T mice could not be detected between the light and dark phases of the day, suggesting that this diurnal organization may be disrupted in the A53T mice.

## 4. Materials and Methods

### 4.1. Animals and Ethics Statement

All experimental procedures in this study were approved by the University of Minnesota Animal Care and Use Committee (1706-34859A 06.02.2020). Mice were maintained on a 12 h light/dark cycle with chow and water ad libitum. Adult male C57BL/6J (WT), A53T (Hualpha-Syn (A53T) transgenic line), orx-Cre and orx-Cre/A53T animals were used for this study. The orx-Cre mice were initially obtained from Prof. Takeshi Sakurai (Kanazawa University, Kanazawa Japan) and bred on the C57BL/6J background in our colony. Generation and initial phenotyping of heterozygous orx-Cre and wild type mice was conducted, and has been described previously [31,92]. The A53T mice were obtained from the Jackson Laboratory (ME, US) and bred on a C57BL/6J background in our colony. Heterozygous A53T mice were generated and characterized as described previously [40]. The orx-Cre/A53T mice were generated by crossing orx-Cre positive females and A53T positive males.

### 4.2. Food Intake, Body Mass and Composition

For the phenotyping study, food intake data were obtained by measuring daily food intake for 3 consecutive days during the pre-SPA and EE analysis habituation phase. For the chemogenetic studies, food intake was measured during SPA and EE analysis. Individual body weights were taken on all mice using an electronic balance (Ohaus, Parsippany, NJ, USA). Body composition was estimated by measuring lean and fat mass using magnetic resonance imaging (Echo MRI 3-in-1, Echo Medical System, Houston, TX, USA).

### 4.3. Open Field Test (OFT)

An opaque, white acrylic arena (50 × 50 × 25 cm) was used for this experiment. A video camera was installed 40 cm above the center of the maze. The camera was connected to a computer and ANY-maze software (San Diego Instruments, San Diego, CA, USA) was used to track and analyze the movement in real-time mode. The light intensity was set to 250 lux measured at the arena level. Animals were i.p. injected either with saline or 3 mg/kg of CNO dissolved in saline 30 min prior to the test. Mice were placed in the middle of the arena and allowed to freely explore the arena for 10 min. Total distance traveled was recorded and analyzed.

### 4.4. SPA and EE Analysis

The Comprehensive Laboratory Animal Monitoring System (CLAMS™, Columbus Instruments, Columbus, OH, USA) is a set of live-in cages for automated, non-invasive calorimetry assessment. First, for the phenotyping study, mice were habituated to single housing for 3 consecutive days. Body mass and composition were measured, and animals were placed in CLAMS cages while oxygen consumption and CO2 production were recorded for 48 h (the first 24 h were considered the habituation period while the second 24 h were used in the analysis). Respiratory exchange ratio (RER) was calculated as the volume of CO2 expired versus volume of oxygen consumed (VCO2/VO2) and used to calculate EE using a formula provided by the manufacturer. Energy expenditure (EE) was expressed as Kcal/h and analyzed with lean body mass as a covariate in an ANCOVA model. Spontaneous physical activity (SPA) was assessed by continuous measurement of ambulatory activity and total movement detected by infrared beam breaks in the X and Y-axes. For the chemogenetic studies habituation was not performed since animals were already housed individually during the recovery period, following the stereotaxic injection surgeries.

### 4.5. Viral Injections and CNO Treatment

Animals were anesthetized with an isofluorane mixture (3% for induction, and 1.5% for maintenance) and placed in a stereotactic apparatus (Kopf Instruments). DREADD targeting was achieved by stereotaxic injection of a Cre-dependent AAV vector expressing a double-floxed inverted open reading frame (DIO) around the DREADD transcript and a fluorescent tag (mCherry). Vectors (AddGene, Watertown, MA, USA) were injected into the LH (AP: −1.8/DV: −5.5/ML: ±0.9 mm from bregma; 333 nL/5 min) [93] of orx-Cre or orx-Cre/A53T mice. Control groups were injected with pAAV-hSyn-DIO-mCherry (AAV8, 2.1 × 1013 GC/mL) (cDREADD). Inhibitory neuromodulation was achieved via Gq-coupled pAAV-hSyn-DIO-hM4D(Gq)-mCherry (AAV8, 2.5 × 1013 GC/mL) (iDREADD). Animals recovered from the surgery for four weeks and were randomly assigned to experimental groups prior to testing.

Water-soluble CNO dihydrochloride was obtained from HelloBio, UK. For behavioral tests, 3 mg/kg of CNO in saline or saline was injected via a small gauge (32) syringe once 30 min prior to testing. For indirect calorimetry and SPA studies, animals were given CNO dissolved in drinking water (0.25 mg/mL) or drinking water [94]. This was necessary to leave unbroken the gas-exchange seal of the indirect calorimetry units.

### 4.6. Immunohistochemistry

Mice were perfused intracardially with ice-cold saline, followed by 20 mL of 4% paraformaldehyde (PFA) in PBS (phosphate buffered saline). Brains were harvested and post-fixed in 4% PFA/PBS overnight at 4 °C, followed by 30% (*w*/*v*) sucrose in PBS solution at 4 °C until the brains sank. The brains were imbedded in OCT (Optimal Cutting Temperature Compound; Sakura, Torrance, CA, USA), frozen in dry ice cooled ethanol, and then immediately cut. Coronal brain sections were collected and stored in cryoprotectant (30% (*w*/*v*) sucrose, 30% (*v*/*v*) ethylene glycol, 1% (*w*/*v*) PVP-40 in PB). Brain sections were washed six times for five min with PBS (0.1 M PBS, pH 7.4). After washing, sections were incubated with 5% normal horse serum in PBST for two hours at room temperature. After washing three times in PBST (0.01% Tween in PBS), the sections were incubated with primary antibodies (rabbit anti-p-α syn (Alpha-synuclein (phospho S129)), Abcam, Cambridge, MA, USA; rabbit anti-GFAP (glial fibrillary acidic protein), Abcam, Cambridge, MA, USA; guinea pig anti-IBA1 (ionized calcium-binding adaptor molecule 1), Novus Biologicals, Littleton, CO USA,; goat anti-orexin A, Santa Cruz, Dallas, TX, USA; mouse anti-orexin A, Santa Cruz, Dallas, TX, USA; guinea pig anti-GAD 65 (glutamate decarboxylase 65), Synaptic Systems, Göttingen, Germany; rabbit anti-synaptophysin, Abcam, Cambridge, MA, USA; guinea pig anti NeuN (hexaribonucleotide Binding protein-3), Millipore, Burlington, MA, USA; 1:1000) overnight at RT on a platform shaker. Brain sections were washed in PBST four times for ten min after primary antibody incubation and incubated with secondary antibodies conjugated with Alexa Fluor dyes (donkey anti-mouse, donkey anti-rabbit, donkey anti-goat, donkey anti-guinea pig; 1:500, Invitrogen, Carlsbad, CA, USA). Brain sections were then washed four times for ten min in PBST and then mounted with ProLong Gold mounting media (Invitrogen, Carlsbad, CA, USA).

### 4.7. Immunofluorescence Imaging and Image Analysis

Immunofluorescence images for GFAP and IBA1 densitometry and IBA1 positive cell density experiments were captured using the Nikon Eclipse NI-E microscope (Nikon, Tokyo, Japan), with a monochrome Nikon Black & White camera DS-QiMc (Nikon, Tokyo Japan). Each fluorochrome is represented as a pseudo-color in the images. For quantification of GFAP and IBA1, every 6th coronal section from −0.94 to −2.18 bregma [93] (five sections total) containing the LH region was collected and analyzed. Optical density was determined with image analysis software (Image J, National Institutes of Health) by measuring the mean gray value of the LH (20× magnification, two images per area, ten in total). For IBA cell density, Z-stack images (5 µm step) were captured using 20× magnification. The IBA positive cell density in the LH region was determined using Image J by counting the positive cells in two areas of the LH and of every 6th section (eight in total) and divided by the ROI area. To determine the percent of orexin A positive cells containing p-α syn, every 6th coronal section from −0.94 to −2.18 bregma [93] (five in total) was analyzed. Images at 40×, Z-stack (5 µm step) were captured using a Nikon C2 Automated Upright Widefield and Confocal Microscope (Nikon, Tokyo, Japan).

### 4.8. Unbiased Stereology

Unbiased stereology analyses with an optical fractionator probe within the Stereo Investigator 11.1.2 software (MBF Bioscience, Williston, VT, USA) were used to quantify the number of orexin A positive cell populations in LH. Sections were cut at 40 μm to allow for an 18 μm dissector height within each section after dehydration and mounting. Systematic sampling of every 3rd section was performed through the orexin field beginning at bregma −0.94 and finishing at −2.18 [93], with the first sampled set of sections chosen at random. Sections were imaged using an Axio Imager M2 fluorescence microscope (Zeiss, Oberkochen, Germany). Orexin field boundaries were used to outline contours at 5× magnification. Cells were counted using a randomly positioned grid system controlled by Stereo Investigator in a previously defined region in all optical planes. Guard zones were set at 10% of the section thickness to account for damage during the staining procedure. The grid size was set to 100 × 100 μm and the counting frame to 80 × 80 μm. Counting was performed on 63× magnification (oil). The average coefficient of error (CE, m = 1) ratio for all of the mice imaged was 0.085. On average, approximately 260 neurons were counted throughout the entire orexin field of each mouse to give an acceptable coefficient of error (CE) (Gunderson method) of 0.085 using the smoothness factor m = 1. The CE provides a means to estimate sampling precision, which is independent of natural biological variance. As the value approaches 0, the uncertainty in the estimate precision reduces. CE < 0.1 is deemed acceptable within the field of stereology. Cells were only counted if they touched the inclusion border or did not touch the exclusion border of the sampling grid.

### 4.9. Quantification of Pre-Synaptic Inhibitory Terminals

Quantification of inhibitory pre-synaptic terminals was performed using an immunocytochemistry-based assay and puncta analyzer ImageJ plugin [95,96]. Three independent coronal brain sections per each mouse (16 μm thick, 3 images per section), containing the LH (bregma from −0.94 to −2.18) [93] were stained with GAD65 (glutamate decarboxylase, 65 kDa isoform localized predominantly in synaptic terminals), synaptophysin (presynaptic protein associated with small synaptic vesicles), and orexin A. The 5-µm-thick confocal scans (optical section depth 0.33 m, 15 sections/scan) of the mPFC were performed at 60× magnification on a Nikon C2 Automated Upright Widefield and Confocal Microscope (Nikon, Tokyo, Japan). Maximum projections of 3 consecutive optical sections (corresponding to 1 µm total depth) were generated. The Puncta Analyzer Plugin for ImageJ, image analysis software (National Institutes of Health) was used to count the number of co-localized pre-synaptic markers. Details of the quantification method using puncta analyzer plugin were given by Ippolito and Eroglu (2010).

### 4.10. Statistical Analyses

All data were analyzed using either Prism 6.0 (GraphPad Software, San Diego, CA, USA) or SPSS (IBM, Armonk, NY, USA). Statistical analyses of phenotyping data were performed using a two-way ANOVA followed by Sidak’s post hoc analysis. Densitometry, cell (IBA1), and pre-synaptic terminals count data were analyzed using Student’s *t*-test. Statistical analyses of DREADD study data were performed using a one-way ANOVA followed by Tukey’s post hoc analysis. Densitometry and IBA1 cell count data for phenotyping study were analyzed using one-way ANOVA followed by Tukey’s post hoc analysis. Unbiased stereology data were analyzed using two-way ANOVA followed by Sidak’s post hoc analysis. The EE data were analyzed using an ANCOVA model with lean body mass as a covariate followed by Sidak’s post hoc test.

### 4.11. Experimental Design and Exclusion Criteria

The initial phenotyping study was performed on male 3-, 5-, 7-, 9- and 11-month-old WT and A53T mice. Mice were first introduced to the OFT (*n* = 9/group) and 7 days later to EE and SPA analysis (*n* = 6/group). Three days following testing in the CLAMS, the animals were sacrificed, and their brains were collected for analysis. Seven-month-old mice (*n* = 5/group) were used for IHC analysis and for the unbiased stereology analysis (*n* = 4/group).

Giving CNO in the water for the DREADD studies in the indirect calorimeter meant that we needed to determine whether CNO itself affected drinking of water, since unequal water intake would complicate data interpretation. To test CNO effects on water consumption, 7-month-old orx-Cre were used. Animals were subjected to viral intracranial injections containing cDREADD. Four weeks following surgery animals were introduced to either water (orx-Cre cDREADD water) or CNO dissolved in drinking water (0.25 mg/mL) (orx-Cre cDREADD CNO) (*n* = 6/group). Water intake was measured daily for 3 days.

To test if CNO affects exploratory locomotion, 7-month-old orx-Cre mice were used. Animals were subjected to viral intracranial injections containing cDREADD. Four weeks following surgery, animals were introduced to the OFT (*n* = 9/group). The mice were injected with either saline (orx-Cre cDREADD saline) or CNO (3 mg/kg) dissolved in saline (orx-Cre cDREADD CNO) 30 min prior to the behavioral test.

To test if CNO affects EE and SPA, 7-month-old orx-Cre mice were used. Animals were subjected to viral intracranial injections containing cDREADD. Four weeks following surgery animals were introduced to the indirect calorimetry chambers for SPA and EE analysis. The mice were given either water (orx-Cre cDREADD water) or CNO dissolved in drinking water (0.25 mg/mL) (orx-Cre cDREADD CNO) (*n* = 6/group).

The chemogenetic study was performed in male 7-month-old orx-Cre and orx-Cre/A53T animals given viral intracranial injections containing either cDREADD or iDREADD. After a 4-week recovery period, animals were introduced to the OFT (*n* = 10/group). In this test, mice were injected with 3 mg/kg of CNO dissolved in saline 30 min prior to the test (orx-Cre cDREADD CNO; orx-Cre/A53T cDREADD CNO; orx-Cre/A53T iDREADD CNO). Seven days following the OFT, mice were introduced to SPA and EE analysis (*n* = 6/group). The mice were given CNO dissolved in drinking water (0.25 mg/mL) (orx-Cre cDREADD saline; orx-Cre cDREADD CNO).

All animals used in the chemogenetic study were perfused, and their brains were collected for injection placement confirmation. Coronal sections containing LH from −0.94 to −2.18 bregma were collected and analyzed. Animals were excluded from the experiment if post hoc histological analyses showed inaccurate viral injection placement. For DREADD expression confirmation, the brains from all animals were analyzed. Every sixth coronal section containing LH from −0.94 to −2.18 bregma (*n* = 5/group) was stained for orexin A and co-localization with mCherry was analyzed.

## 5. Conclusions

This study provided several novel findings. Firstly, age-dependent exploratory locomotion, SPA, and EE disturbances are present in the A53T model of PD. Further, the observed SPA data suggest a possible loss of sleep–wake rhythm in A53T mice, an observation that deserves more detailed investigation. The observed changes were accompanied by PD-associated pathology in the LH, which was represented by early p-α syn intra-cellular accumulations in orexin neurons, astrogliosis and inflammation, loss of inhibitory pre-synaptic terminals, and late orexin neuronal loss. Finally, the observed PD pathology-associated disturbances in exploratory locomotion, SPA, and EE were ameliorated by orexin neuronal inhibition using DREADDs. These findings propose A53T mice as an interesting model for hypothalamic-regulated physiological function disturbances in PD and highlight the role of orexin neurons in PD-associated pathology.

## Figures and Tables

**Figure 1 ijms-22-00795-f001:**
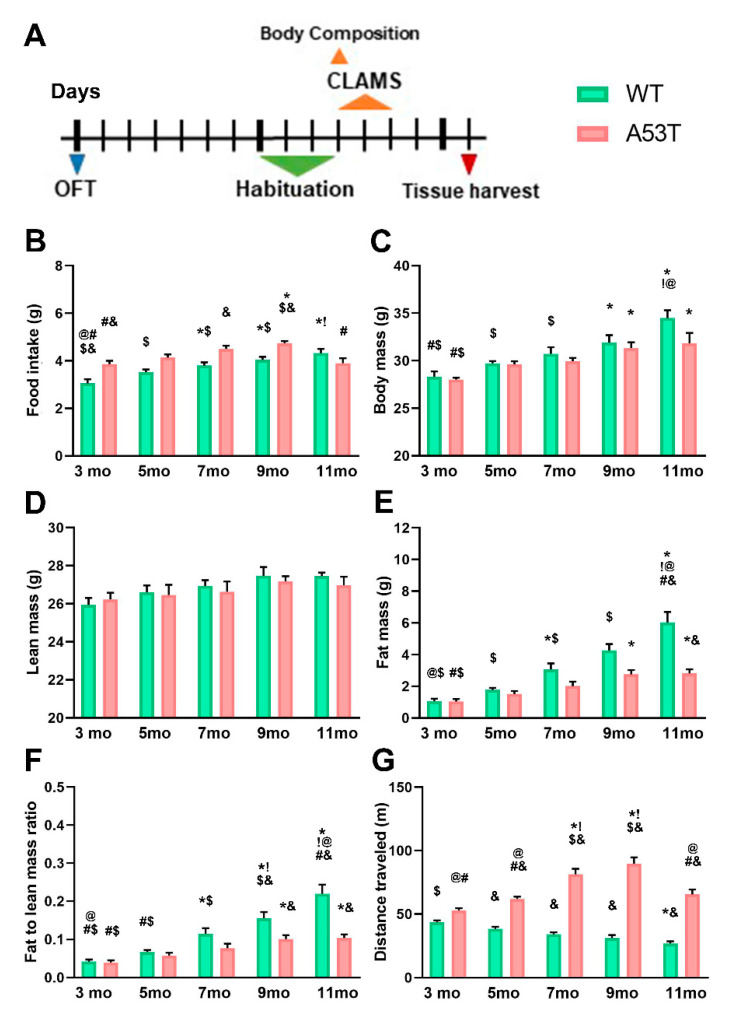
Body composition and exploratory locomotion in 3-, 5-, 7-, 9-, and 11-month-old WT and A53T mice. A timeline of the experimental procedures (**A**). The open field test (OFT) was followed by food intake, body composition, spontaneous physical activity (SPA), and energy expenditure (EE) measurements. Aging induces increase in food intake in both WT and A53T mice, except in 11-month-old A53T mice. Compared to WT animals, A53T mice tend to consume more food (**B**). The increase in body weight is observable at both WT and A53T mice as they age (**C**). Differences in lean mass were not observed (**D**) however, age-associated increase in fat mass is significantly lower in A53T mice compared to WT (**E**) mice leading to significant differences in the fat to lean mass ratio in 9- and 11-month-old mice (**F**). As expected, WT mice showed an age-related reduction in distance covered in the OFT. Compared to WT mice, A53T mice covered more distance in the OFT and had an age-associated increase in distance covered in the OFT (**G**). (Body composition: *n* = 6/group; OFT, food intake: *n* = 9/group; *p* < 0.05 * vs. 3 mo; ! vs. 5 mo; @ vs. 7 mo; # vs. 9 mo; $ vs. 11 mo; & vs. group of the same age and other genotype).

**Figure 2 ijms-22-00795-f002:**
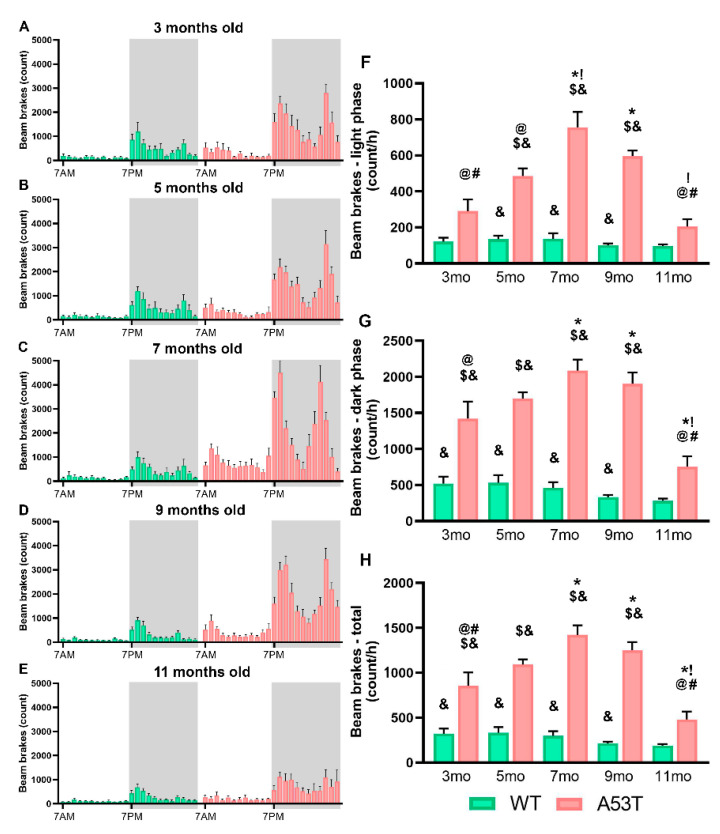
Spontaneous physical activity (SPA) in 3-, 5-, 7-, 9-, and 11-month-old WT and A53T mice. SPA measured per hour during the 24 h period shown by the number of beam breaks for 3-month-old (**A**), 5-month-old (**B**), 7-month-old (**C**), 9-month-old (**D**), 11-month-old (**E**) WT and A53T mice. The white area represents the light phase while the gray area represents the dark phase. The A53T mice show increased SPA compared to WT mice during the light, inactive phase at 7 and 9 months of age (**F**). Increased SPA is also observed in A35T mice during the dark, active phase (**G**). Similar changes were observed in total daily SPA (**H**). In A53T mice, SPA increased with age and reached a peak at 7–9 months of age. (*n* = 6/group; *p* < 0.05 * vs. 3 mo; ! vs. 5 mo; @ vs. 7 mo; # vs. 9 mo; $ vs. 11 mo; & vs. group of the same age and other genotype).

**Figure 3 ijms-22-00795-f003:**
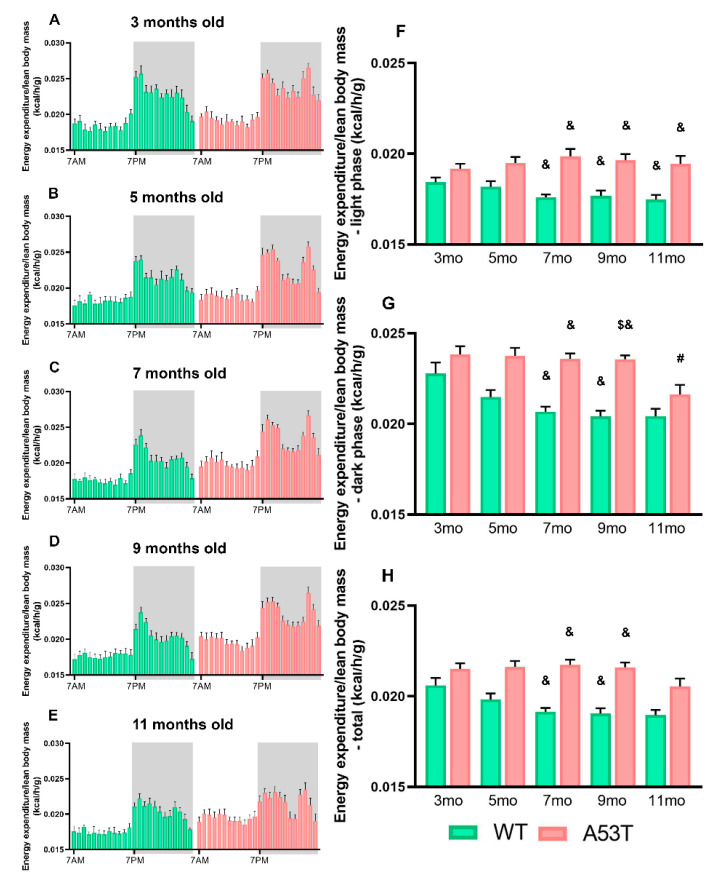
Energy expenditure (EE) in 3-, 5-, 7-, 9-, and 11-month-old WT and A53T mice. EE measured per hour during the 24 h period shown as the EE/h/lean body mass (kcal/h/g) for 3-month-old (**A**), 5-month-old (**B**), 7-month-old (**C**), 9-month-old (**D**), 11-month-old (**E**) WT and A53T mice. The white area represents the light phase while the gray area represents the dark phase. F: Light, inactive phase; differences in EE were observed between 7-, 9-, and 11-month-old WT and A53T mice. (**G**) Dark, active phase, differences in EE were observed between 7- and 9-month-old WT and A53T mice. A significant reduction in EE was observed between 9- and 11-month-old A53T mice. (**H**) Total daily EE differences in EE were observed between 7- and 9-month-old WT and A53T mice (*n* = 6/group; *p* <0.05 # vs. 9 mo; $ vs. 11 mo; & vs. group of the same age and other genotype).

**Figure 4 ijms-22-00795-f004:**
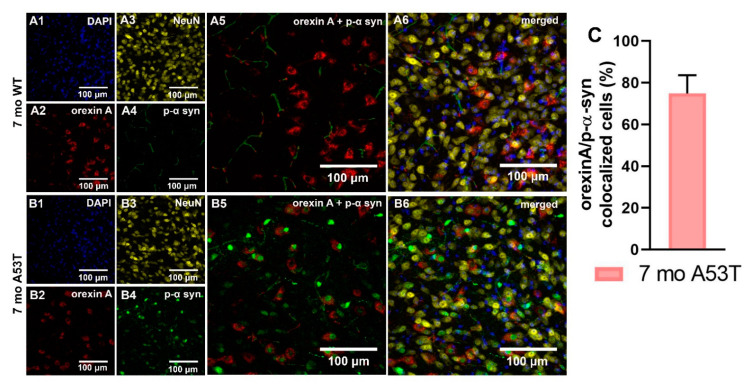
Quantification and localization of orexin A neuronal α-syn accumulations. Representative IF microphotographs of the DAPI in blue (**A1**,**B1**), orexin A in red (**A2**,**B2**), NeuN in yellow (**A3**,**B3**), p-α-syn in green (**A4**,**B4**), and merged images (**A5**,**A6**,**B5**,**B6**) in 7 mo WT (**A**) and A53T (**B**) mice showing the presence of the p-α-syn in the orexin neurons. (**C**) Percent of orexin neurons expressing p-α-syn defined as orexin A/p-α-syn co-localized cells in 7 mo A53T mice.

**Figure 5 ijms-22-00795-f005:**
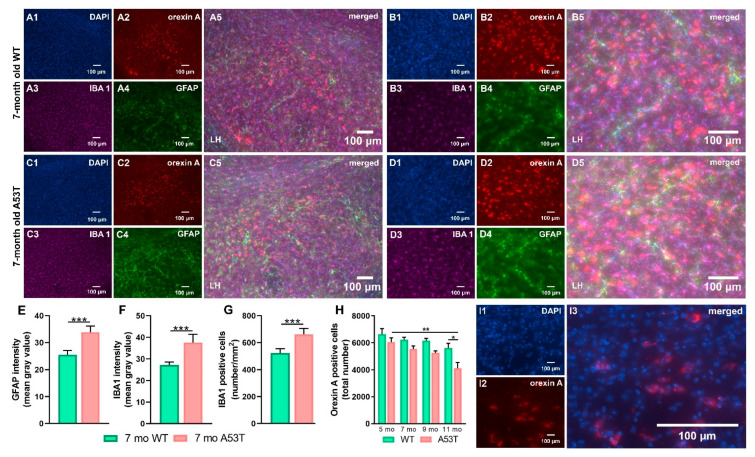
Expression of GFAP and IBA1 and number of the orexin A positive cells in the lateral hypothalamus (LH) of WT and A53T mice. Representative IF microphotographs of the DAPI in blue (**A1**,**B1**,**C1**,**D1**), orexin A in red (**A2**,**B2**,**C2**,**D2**), IBA1 in purple (**A3**,**B3**,**C3**,**D3**), GFAP in green (**A4**,**B4**,**C4**,**D4**), and the merged images (**A5**,**B5**,**C5**,**D5**) in 7 mo WT mice (**A**) low magnification; (**B**) high magnification and A53T mice (**C**) low magnification; (**D**) high magnification. Image J was used to quantify the intensity of GFAP and IBA1 staining and density of IBA1 positive cells. Increased expression of the GFAP (**E**) and IBA1 (**F**) was observed in A53T mice compared to WT mice. The A53T mice showed increased density of IBA1 positive cells (**G**). Unbiased stereology analysis showed reduced number of the orexin A positive neurons between 11 mo WT and A53T mice as well as age-associated loss of orexin neurons in A53T mice (**H**). Representative high magnification (63× oil) micrographs of the DAPI in blue (**I1**), orexin A in red (**I2**) and merged image (**I3**) used for unbiased stereology analysis. Densitometry and IBA1 positive cell numbers: *n* = 5/group; Student’s *t*-test; Unbiased stereology: *n* = 4/group; Two-way ANOVA, Sidak’s; * *p* < 0.05, ** *p* < 0.01, *** *p* < 0.005).

**Figure 6 ijms-22-00795-f006:**
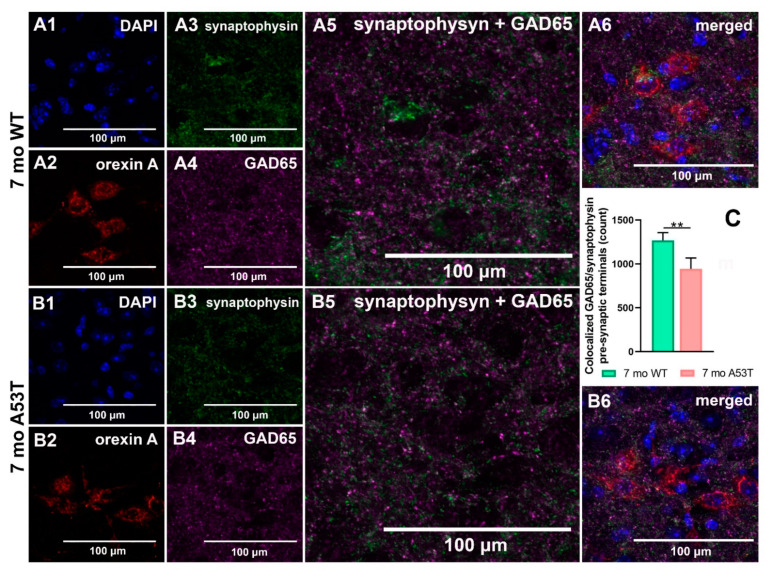
Quantification of inhibitory pre-synaptic terminals in the LH of 7-month-old WT and A53T mice. Representative high magnification IF microphotographs of the GAD65 and synaptophysin and merged images of the LH of 7 mo WT mice (**A**) and A53T mice (**B**). DAPI in blue (**A1**,**B1**), orexin A in red (**A2**,**B2**), synaptophysin in green (**A3**,**B3**) and GAD 65 in purple (**A4**,**B4**). Merged images: synaptophysin and GAD65 (**A5**,**B5**); DAPI, orexin A, synaptophysin and GAD65 (**A6**,**B6**). Immunocytochemistry-based assay showed reduced number of co-localized GAD65/synaptophysin pre-synaptic terminals in 7 mo A53T mice compared to WT littermates (**C**). Student’s *t*-test, *n* = 5/group; ** *p* < 0.01.

**Figure 7 ijms-22-00795-f007:**
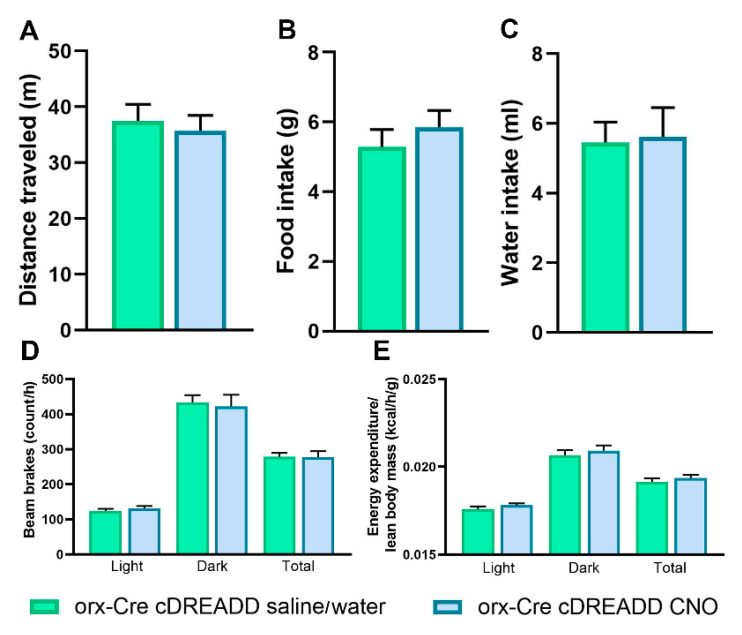
Effect of CNO on exploratory locomotion, water intake, SPA, and EE in 7-month-old male orx-Cre mice. Orx-Cre mice were subjected to intracranial injections of virus containing the control DREADD construct. For CNO effects on exploratory locomotion, animals were injected either with saline or CNO (3 mg/kg). Compared to saline-treated mice, CNO-treated mice showed no difference in distance traveled in the OFT (**A**). For CNO effects on water consumption, SPA and EE animals were either introduced to drinking water or CNO dissolved in drinking water (0.25 mg/mL). CNO dissolved in drinking water did not affect food intake (**B**) or water consumption (**C**). CNO treatment did not affect SPA or EE (**D**,**E**). OFT: *n* = 10/group; food intake, water intake, SPA, EE: *n* = 6/group; Student’s *t*-test.

**Figure 8 ijms-22-00795-f008:**
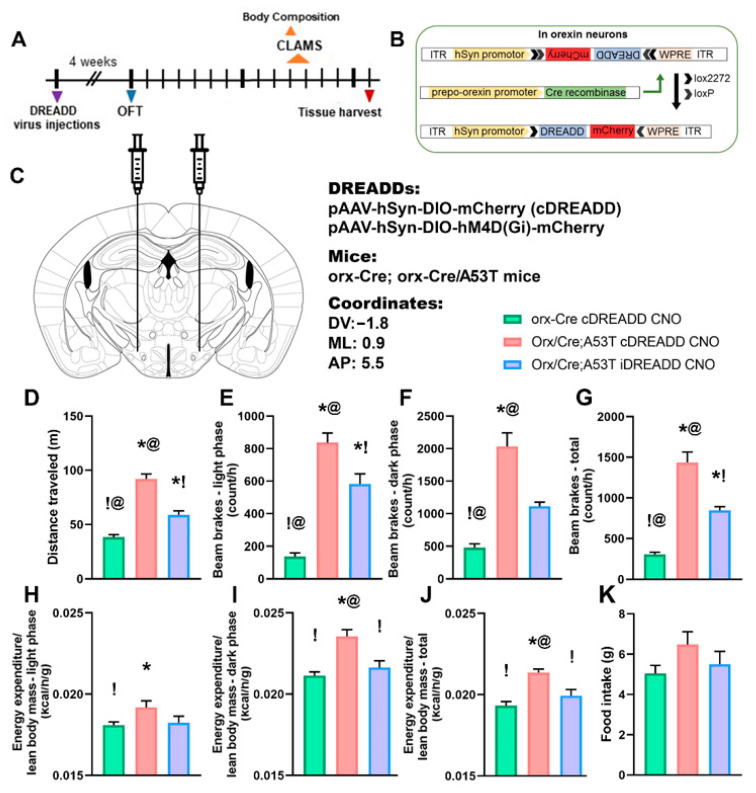
Chemogenetic inhibition of orexin neurons reduces exploratory locomotion, SPA, and EE in 7-month-old orx-Cre/A53T mice. The timeline of the experimental procedures (**A**). Orx-Cre mice received virus containing control DREADD construct, while orx-Cre/A53T mice received virus containing either control DREADD or inhibitory DREADD construct. After four weeks of recovery time the OFT test was performed. One week following the OFT, body composition analysis was performed, and mice were introduced to CLAMS for SPA and EE measurements. Three days following CLAMS, mice were perfused, and brains were collected. Schematic diagram of AAV vector encoding DREADD-mCherry driven by human synapsin promoter (hSyn) promoter sequence and flanked by dual flox sites for recombination in the presence of Cre-recombinase. (**B**) Cre expression in orx-Cre mice is driven by the prepro-orexin promoter. Schematic representation of DREADD virus injection site within the lateral hypothalamus (LH). (**C**) DREADD-virus constructs were injected bilaterally (333 nL/5 min). (**D**) Chemogenetic inhibition of orexin neuronal activity reduced exploratory locomotion in 7-month-old orx-Cre/A53T mice. (**E**) DREADD induced inhibition of orexin neurons reduced SPA in both light and dark phase of the day, as well as total SPA (**E**–**G**). Chemogenetic inhibitory intervention on orexin neurons did not affect light phase EE (**H**). However, inhibition of orexin neurons reduced EE in both dark phase and total EE (**I**,**J**). Food intake was not affected by chemogenetic intervention (**K**). (OFT: *n* = 10/group; food intake, SPA, EE: *n* = 6/group; OFT, SPA: One-way ANOVA, Tuckey; EE: ANCOVA, Sidak’s; *p* < 0.05 * vs. Orx-Cre cDREADD CNO; ! vs.Orx/Cre;A53T cDREADD CNO; @ vs. Orx/Cre;A53T iDREADD CNO).

**Figure 9 ijms-22-00795-f009:**
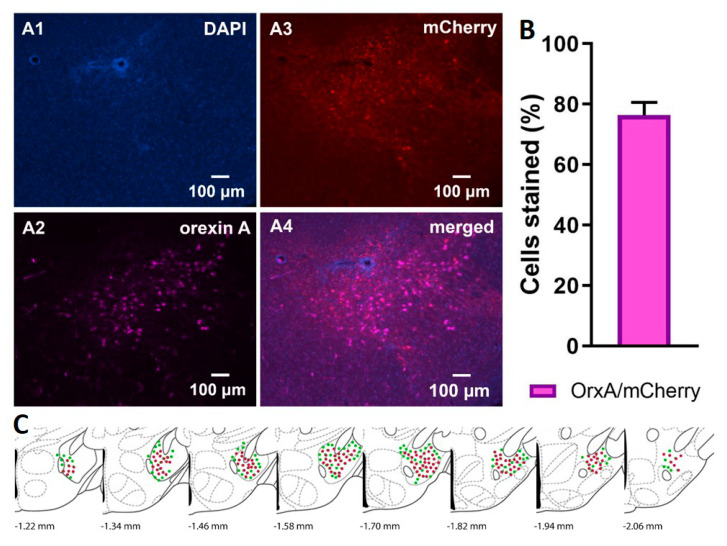
DREADD expression confirmation. Representative images displaying viral expression of DREADDs in the LH (**A**), DAPI in blue (**A1**), orexin A positive neurons in purple (**A2**), mCherry positive neurons in red (**A3**), and merged images (**A4**). The percentage of OrxA/mCherry co-localized cells (**B**). Schematic drawings displaying the spread of viral expression along the LH; green orexin A expressing cells, red mCherry expressing cells (**C**). (*n* = 5/group).

**Table 1 ijms-22-00795-t001:** Body composition and exploratory locomotion in 3-, 5-, 7-, 9-, and 11-month-old WT and A53T mice (*p* < 0.05 * vs. 3mo; ! vs. 5 mo; @ vs. 7 mo; # vs. 9 mo; $ vs. 11 mo; & vs. group of the same age and other genotype).

			Body Composition				Exploratory Locomotion
Genotype	Age (Months)	Food Intake (g)	Body Mass (g)	Lean Mass (g)	Fat Mass (g)	Fat to Lean Mass Ratio	Distance Traveled (m)
WT	3	3.05 ± 0.17 @#$&	28.30 ± 0.57 #$	25.94 ± 0.37	1.07 ± 0.14 @$	0.0413 ± 0.0058 @#$	43.38 ± 1.56 $
	5	3.52 ± 0.12 $	29.72 ± 0.24 $	26.61 ± 0.36	1.78 ± 0.13 $	0.0670 ± 0.0055 #$	38.26 ± 1.73 &
	7	3.80 ± 0.14 *&	30.72 ± 0.70 $	26.93 ± 0.30	3.07 ± 0.37 *$	0.1142 ± 0.0146 *$	34.06 ± 1.45 &
	9	4.04 ± 0.13 *&	31.90 ± 0.80 *	27.47 ± 0.46	4.25 ± 0.42 $	0.1552 ± 0.0162 *!$&	31.26 ± 2.15 &
	11	4.32 ± 0.18 *!	34.47 ± 0.85 *!@	27.45 ± 0.19	6.02 ± 0.67 *!@#&	0.2192 ± 0.0242 *!@#&	27.08 ± 1.54 *&
A53T	3	3.86 ± 0.14 #&	28.02 ± 0.18 #$	26.22 ± 0.36	1.03 ± 0.16 #$	0.0392 ± 0.0060 #$	52.63 ± 1.96 @#
	5	4.14 ± 0.13	29.60 ± 0.35	26.46 ± 0.53	1.51 ± 0.19	0.0576 ± 0.0078	61.87 ± 1.84 @#&
	7	4.50 ± 0.14 &	30.72 ± 0.35	26.63 ± 0.54	2.01 ± 0.28	0.0768 ± 0.0122	81.34 ± 4.37 *!$&
	9	4.74 ± 0.09 *$&	31.32 ± 0.62 *	27.41 ± 0.28	2.75 ± 0.27 *	0.1011 ± 0.0096 *&	89.70 ± 5.02 *!$&
	11	3.90 ± 0.21 #	31.82 ± 1.10 *	26.98 ± 0.44	2.81 ± 0.26 *&	0.1039 ± 0.0089 *&	65.40 ± 3.91 @#&

**Table 2 ijms-22-00795-t002:** Spontaneous physical activity (SPA) and energy expenditure (EE) in 3-, 5-, 7-, 9-, and 11-month-old WT and A53T mice (*p* < 0.05 * vs. 3 mo; ! vs. 5 mo; @ vs. 7 mo; # vs. 9 mo, $ vs. 11 mo; & vs. group of the same age and other genotype).

		Spontaneous Physical Activity (SPA)			Energy Expenditure (EE)		
Genotype	Age (months)	Beam Breaks—Light Phase (count/h)	Beam Breaks—Dark Phase (count/h)	Beam Breaks—Total (count/h)	Energy Expenditure/Lean Body Mass—Light Phase (kcal/h/g)	Energy Expenditure/Lean Body Mass—Dark Phase (kcal/h/g)	Energy Expenditure/Lean Body Mass—Total (kcal/h/g)
WT	3	121.62 ± 21.60	522.03 ± 96.71 &	321.82 ± 57.91 &	0.01842 ± 0.00027	0.02277 ± 0.00061	0.02060 ± 0.00042
	5	134.39 ± 20.41 &	532.01 ± 105.72 &	333.20 ± 62.34 &	0.01818 ± 0.00031	0.02148 ± 0.00039	0.01983 ± 0.00032
	7	136.04 ± 31.71 &	462.14 ± 78.21 &	299.09 ± 51.41 &	0.01760 ± 0.00016 &	0.02067 ± 0.00028 &	0.01913 ± 0.00022 &
	9	101.11 ± 9.83 &	332.15 ± 31.23 &	216.63 ± 18.07 &	0.01768 ± 0.00029 &	0.02041 ± 0.00032 &	0.01905 ± 0.00029 &
	11	96.27 ± 9.68	284.67 ± 28.33	190.47 ± 16.65	0.01748 ± 0.00024 &	0.02043 ± 0.00042	0.01896 ± 0.00029
A53T	3	291.06 ± 64.15 @#	1421.47 ± 237.51 @$&	856.27 ± 148.58 @#$&	0.1918 ± 0.0027	0.02383 ± 0.00045	0.02151 ± 0.00032
	5	485.76 ± 42.57 @$&	1700.42 ± 85.64 $&	1093.09 ± 57.36 $&	0.01949 ± 0.0033	0.02375 ± 0.00044	0.02162 ± 0.00032
	7	756.28 ± 85.30 *!$&	2087.91 ± 152.22 *$&	1422.09 ± 107.65 *$&	0.01986 ± 0.0040 &	0.02359 ± 0.00030 &	0.02173 ± 0.00029 &
	9	595.45 ± 31.56 *$&	1905.05 ± 157.43 *$&	1250.25 ± 92.00 *$&	0.01963 ± 0.0035 &	0.02357 ± 0.00021 $&	0.02160 ± 0.00026 &
	11	204.73 ± 41.09 !@#	755.40 ± 143.23 *!@#	480.07 ± 89.59 *!@#	0.01944 ± 0.0045 &	0.02163 ± 0.00052 #	0.02053 ± 0.00044

**Table 3 ijms-22-00795-t003:** Effects of chemogenetic inhibition of orexin neurons on exploratory locomotion, SPA, EE, and food intake in 7-month-old mice. (*p* < 0.05 * vs. Orx-Cre cDREADD CNO; ! vs. Orx/Cre;A53T cDREADD CNO; @ vs. Orx/Cre;A53T iDREADD CNO).

	Exploratory Locomotion	Spontaneous Physical Activity (SPA)		
Genotype	Distance traveled (m)	Beam breaks—light phase (count)	Beam breaks—dark phase (count)	Beam breaks—total (count)
Orx-Cre cDREADD CNO	38.37 ± 2.30 !@	136.69 ± 24.12 !@	480.69 ± 55.24 !@	308.69 ± 27.26 !@
Orx/Cre;A53T cDREADD CNO	92.11 ± 4.62 *@	837.18 ± 62.52 *@	2035.29 ± 207.31 *@	1436.24 ± 130.06 *@
Orx/Cre;A53T iDREADD CNO	58.78 ± 3.80 *!	582.85 ± 62.97 *!	1113.04 ± 64.17 *!	847.94 ± 44.36 *!
	Energy expenditure (EE)			
Genotype	Energy expenditure/lean body mass—light phase (kcal/h/g)	Energy expenditure/lean body mass—dark phase (kcal/h/g)	Energy expenditure/lean body mass—totlal (kcal/h/g)	Food intake (g)
Orx-Cre cDREADD CNO	0.01809 ± 0.00020 !	0.02113 ± 0.00024 !	0.01932 ± 0.00026 !	5.05 ± 0.40
Orx/Cre;A53T cDREADD CNO	0.01919 ± 0.00040 *	0.02356 ± 0.00042 *@	0.02136 ± 0.00022 *@	6.48 ± 0.63
Orx/Cre;A53T iDREADD CNO	0.01824 ± 0.00040	0.02164 ± 0.0041 !	0.01994 ± 0.00038 !	5.50 ± 0.64

## Data Availability

The data presented in this study are available on request from the corresponding author.

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
