# Peer review of "Inhibition of Orexin/Hypocretin Neurons Ameliorates Elevated Physical Activity and Energy Expenditure in the A53T Mouse Model of Parkinson’s Disease"

_ijms, 2021, doi:10.3390/ijms22020795_

Round 1

Reviewer 1 Report

Manuscript Number - ijms-1065907

Authors - Milos Stanojlovic1 et al

Title: Inhibition of orexin/hypocretin neurons ameliorates elevated

physical activity and energy expenditure in the A53T mouse model of

Parkinson's disease

The authors used A53T mice as an animal model for PD. They explored the role of hypocretin (orexin) in the pathogenesis of PD. Previous works reported that orexin neuronal loss and orexin system impairment is present in Parkinson's disease patients.

This original research work is exciting and has significance for Parkinson’s disease.

This study is based on the hypothesize that PD associated

pathology affects orexin neurons and therefore impairs functions they regulate.

To establish this hypothesis the authors used multiple methods to come with solid results. The results show that (a) behavioral and energy metabolism impairment in Parkinson’s disease mice model, (b) early presence of alpha-synuclein accumulations in orexin neurons, (c) early loss of inhibitory pre-synaptic terminals in lateral hypothalamus, (d) loss of orexin neurons at late stages of the disease and (e) chemogenetics targeting orexin neurons reduce behavioral and metabolism impairment.

The authors have substantially enough data to conclude this work implicates orexin's involvement in early Parkinson’s disease-associated impairment of hypothalamic-regulated physiological functions and highlights the importance of orexin neurons in Parkinson’s disease symptomology.

These findings prove that A53T mice as an interesting model for hypothalamic-regulated physiological function disturbances in PD and highlight the role of orexin neurons in PD-associated pathology.

Major comments

  • The last two paragraphs of the introduction are a mix of methods, results, and conclusions. These paragraphs have to be modified.
  • One of the significant findings is that the early presence of alpha-synuclein accumulations in orexin neurons. In human PD patients, there are practically no alpha-synuclein accumulations in orexin neurons (Thannickal et al. 2007). How do the authors explain this? In human PD, there is also a loss of MCH neurons (Thannickal et al., 2007). In this A53T mice model, the alpha-synuclein accumulations are limited to orexin neurons, or other neurons like MCH have the same phenomenon? This should be at least included in the discussion.

Minor Comments:

  • Page 2, lines 56-57 reference number repeated in the same sentence.
  • Page 2, line 59 Hualpha-Syn change to human alpha -syn
  • Page 2, line 64 reference number [39] and author’s name mentioned.
  • Figure 2 figure legend [F] not mentioned.
  • The histological figures are in low-quality resolution.

Author Response

We would like to thank the reviewer for the useful critique. We followed the reviewer recommendations and corrected the manuscript accordingly. We feel the manuscript is much improved and meets the criteria for publishing.

Reviewer 1

Major comments

Point 1: The last two paragraphs of the introduction are a mix of methods, results, and conclusions. These paragraphs have to be modified.

Response 1: Thank you for this observation. The paragraphs have been rewritten to accommodate the reviewer’s suggestions.

(Lines 69-76): ”Classically, PD is considered a motor disorder driven by dopamine system impairment, but the importance of non-motor symptoms and pathophysiological mechanisms involving different brain regions, neuronal populations and signaling pathways has been recently recognized [47,48]. In this study, we determined the effects of A53T α-syn associated pathology on aspects of energy metabolism, including body composition, exploratory locomotor activity, SPA, EE and LH orexin. Further, we investigated if chemogenetic inhibition of orexin neurons can ameliorate impairments observed in A53T mice.”

Point 2: One of the significant findings is that the early presence of alpha-synuclein accumulations in orexin neurons. In human PD patients, there are practically no alpha-synuclein accumulations in orexin neurons (Thannickal et al. 2007). How do the authors explain this? In human PD, there is also a loss of MCH neurons (Thannickal et al., 2007). In this A53T mice model, the alpha-synuclein accumulations are limited to orexin neurons, or other neurons like MCH have the same phenomenon? This should be at least included in the discussion.

Response 2: We agree and the discussion has been expanded in line with the reviewer’s recommendations.

(Lines 359-372) “The presence of Lewy bodies in the LH of PD patients is established [73,74], but only one study has addressed if α-syn accumulations are observed in orexin neurons [75]. While data from the study by Thannickal et al., 2007, showed no α-syn accumulations in orexin neurons, more sophisticated approaches could yield a different outcome.  Further, alpha-syn accumulations undoubtedly play a role in the pathology of PD, however p-α-syn, insoluble α-syn aggregates, and oligomeric α-syn species are implicated in PD pathophysiology as well [76-78]. Moreover, orexin neurons are not the only LH neuronal population susceptible to neurodegeneration, as the loss of LH melanin-concentrating hormone (MCH) neurons has been shown in PD patients [75]. Both orexin and MHC neurons are closely linked to the dopaminergic system [79,80], and it would be interesting to examine if dopamine system impairment is a common underlying mechanism of neurodegeneration in orexin and MHC neurons, or if different mechanisms exist.”

Minor Comments:

Point 1: Page 2, lines 56-57 reference number repeated in the same sentence.

Response 1: Corrected as reviewer indicated.

Point 2: Page 2, line 59 Hualpha-Syn change to human alpha –syn

Response 2: Corrected as reviewer indicated.

Point 3: Page 2, line 64 reference number [39] and author’s name mentioned.

Response 3: Corrected as reviewer indicated.

Point 4: Figure 2 figure legend [F] not mentioned.

Response 4: Corrected as reviewer indicated.

Point 5: The histological figures are in low-quality resolution.

Response 5: Due to the size of the files, we could not upload original tiff files. Images figures that can be found in the manuscript are downsized JPEG images of lower quality. Nevertheless, we provided original files separately during the submission process.

Reviewer 2 Report

The authors investigated orexin neurons and orexin-regulated functions in the A53T mouse, a transgenic animal model of PD. Pathological alpha-synuclein aggregates were detected in orexin neurons as well as loss of inhibitory pre-synaptic terminals and reduced number of orexin cells in A53T mice. In A53T mice, with aging fat mass decreased and exploratory locomotion, spontaneous physical activity, and energy expenditure increased. Chemogenetic inhibition of orexin neurons mitigated the described behaviors. 

The study is methodologically sound and results are well discussed. In my opinion, there are few minor points to be addressed:

1. In the first sentence of the discussion, dyskinesia should be substituted with hypokinesia. Dyskinesia is not a characteristic of PD but a side effect of treatment with dopamine. 

2. Due to alpha-synuclein accumulation, it would be expected a degeneration of orexin neurons. These were reduced in number, but the remaining orexin neurons were over-activated. The authors hypothesize and demonstrated a loss of inhibitory presynaptic terminals to the lateral hypothalamus orexin field. A further hypothesis is that orexin neuronal loss could be a consequence of its prolonged over-activation. It would be useful if the authors could add to the discussion a part on how these finding and hypothesis could relate to finding in PD patients. E.g., an association betweek PD and narcolepsy has been described.

3. Possible loss of sleep-wake rhythm in A53T mice is a relevant finding. Although it was not the aim of this study and further studies are needed to elucidate this aspect, it should nevertheless be mentioned also in the conclusions in my opinion. 

Author Response

We would like to thank the reviewer for the useful critique. We followed the reviewer recommendations and corrected the manuscript accordingly. We feel the manuscript is much improved and meets the criteria for publishing.

Reviewer 2

Point 1: In the first sentence of the discussion, dyskinesia should be substituted with hypokinesia. Dyskinesia is not a characteristic of PD but a side effect of treatment with dopamine. 

Response 1: Thank you for the comment. The correction has been made.

Point 2: Due to alpha-synuclein accumulation, it would be expected a degeneration of orexin neurons. These were reduced in number, but the remaining orexin neurons were over-activated. The authors hypothesize and demonstrated a loss of inhibitory presynaptic terminals to the lateral hypothalamus orexin field. A further hypothesis is that orexin neuronal loss could be a consequence of its prolonged over-activation. It would be useful if the authors could add to the discussion a part on how these finding and hypothesis could relate to finding in PD patients. E.g., an association betweek PD and narcolepsy has been described.

Response 2: The discussion has been expanded to accommodate the reviewer’s suggestions.

(Lines 381-389) “Interestingly, sleep impairments such as insomnia, fragmented sleep and narcolepsy are common in PD patients [84]. The orexins play a major role in the regulation of sleep [85], and thus it is not surprising that orexin system dysregulation results in various sleep impairments, including destabilization and loss of REM sleep [86–88], more sleep to wake transitions [89] and sleep fragmentation [90]. Perhaps the most interesting link between the orexin system and PD is narcolepsy, which is commonly experienced by PD patients [91]. It is well established that orexin neuronal loss causes narcolepsy [92] and occurs in PD patients [75,93], and thus it is possible that narcolepsy observed in PD is caused by orexin neuronal loss.”

Point 3: Possible loss of sleep-wake rhythm in A53T mice is a relevant finding. Although it was not the aim of this study and further studies are needed to elucidate this aspect, it should nevertheless be mentioned also in the conclusions in my opinion.

Response 3: Thank you for the remark. The discussion and conclusion sections have been expanded as below.

(Lines 603-605) “Further, the observed SPA data suggest a possible loss of sleep-wake rhythm in A53T mice, an observation that deserves more detailed investigation.”